# Water Stress, Cadmium, and Plant Genotype Modulate the Rhizosphere Microbiome of *Pisum sativum* L.

**DOI:** 10.3390/plants11223013

**Published:** 2022-11-08

**Authors:** Arina A. Kichko, Grigory V. Gladkov, Pavel S. Ulianich, Vera I. Safronova, Alexander G. Pinaev, Edgar A. Sekste, Andrey A. Belimov, Evgeny E. Andronov

**Affiliations:** 1All-Russia Research Institute for Agricultural Microbiology, Podbelskogo sh. 3, Pushkin, 196608 Saint-Petersburg, Russia; 2Dokuchaev Soil Science Institute, 119017 Moscow, Russia

**Keywords:** rhizosphere, microbiome, cadmium, drought, water stress, cadmium tolerant pea mutant, SGECd^t^, compositional data

## Abstract

Drought and heavy metals seriously affect plant growth and the biodiversity of the associated rhizosphere microbiomes, which, in turn, could be involved in the adaptation of plants to these environmental stresses. Rhizosphere soil was collected from a three-factor pot experiment, where pea line SGE and its Cd-tolerant mutant SGECd^t^ were cultivated under both optimal and limited water conditions and treated with a toxic Cd concentration. The taxonomic structure of the prokaryotic rhizosphere microbiome was analyzed with the high-throughput sequencing of 16S rRNA amplicon libraries. A permutation test demonstrated statistically significant effects of Cd and water stress but not of pea genotype on the rhizosphere microbiome structure. Phylogenetic isometric log-ratio data transformation identified the taxonomic balances that were affected by abiotic factors and pea genotypes. A small number of significant (log ratio [−3.0:+3.0]) and phylogenetically deep balances characterized water stress, while a larger number of weak (log ratio [−0.8:+0.8]) phylogenetically lower balances described the influence of the plant genotype. Stress caused by cadmium took on an intermediate position. The main conclusion of the study is that the most powerful factor affecting the rhizosphere microbiome was water stress, and the weakest factor was plant genotype since it demonstrated a very weak transformation of the taxonomic structure of rhizosphere microbiomes in terms of alpha diversity indices, beta diversity, and the log ratio values of taxonomic balances.

## 1. Introduction

A rhizosphere is a multifunctional interface between plants and soil that ensures a broad spectrum of functions, including plant nutrition, defense against phytopathogens, and adaptation to abiotic and biotic stresses [1,2,3]. One of the most important constituents of the rhizosphere is the microbiome recruited from the soil and shaped by different plant-related mechanisms, including root exudates and immune reactions of the host plant [4,5]. An important factor that affects the structure of the rhizosphere microbiome is drought [6] and the presence of heavy metals as a native component or anthropogenic pollution [7].

As a part of the soil microbiome, the rhizosphere microbiome is also deeply affected by the same environmental factors as the parental soil microbiome. The order of biotic and abiotic factors that are able to influence soil bacterial communities is led by soil pH, followed by organic carbon quality and quantity, soil O_2_ and redox status, water availability, and lastly, the plant species identity [8,9]. When shifting focus to the rhizosphere, the dynamics of the interaction between the rhizosphere microbiome and common environmental factors become more complicated. The rhizosphere is rich in easily consumable carbon compounds and provides microbes with a wide variety of microniches that moderate external effects [10]. In turn, the rhizosphere microbiome may help the plant tolerate stress factors [11].

Plants themselves have great variability in their resistance to abiotic stress due to diverse phenotypic, genotypic, and biochemical mechanisms, but the rhizosphere microbiome also makes a significant contribution to this resistance [12,13] and the resistance of plants to abiotic stresses in the natural environment should be considered a superposition of their own and rhizosphere microbiome-assisted resistance. There are many reports describing the mechanics of such resistance. The microbiome-assisted resistance of plants to heavy metal stress [14,15,16], including cadmium toxicity, may be conditioned by immobilizing and bioaccumulating this toxicant by representatives of *Proteobacteria*, *Actinobacteria*, and *Firmicutes* [7]. In addition, rhizosphere microbiota make their own contribution to the hyperaccumulation of zinc and cadmium by *Arabidopsis halleri* [17,18]. In this regard, reports about the presence of specific metal-tolerant and plant growth-promoting bacteria (PGPB) that facilitate alleviating metal stress for plants deserve consideration [18,19]. For example, *Burkholderiaceae* representatives are known to produce the enzyme 1-aminocyclopropane-1-carboxylate (ACC) deaminase, which alleviates stress by lowering ethylene concentrations in plants exposed to biotic and abiotic stress, including drought and heavy metal toxicity [20]. This observation suggests that heavy metal and drought tolerance mediated by rhizosphere microorganisms may have similar mechanisms [21]. From the point of view of the diversity of dynamics, it should be noted that despite the fact that the presence of heavy metals has affected the microbiome feature of the rhizosphere in most cases, it has both led [18] and not led [22,23] to increased rhizosphere diversity in microbial communities.

Resistance to water stress is also a result of complex interactions. This includes changes in the plant hormone status and root exudate spectrum, leading to modification of the rhizosphere microbiome. The modified microbiome, in turn, can make its own contribution to plant resistance against drought [6,24,25], for example, by producing osmoprotectants as the plant does [26]. Drought stress led to the modification of rhizosphere community-enriching *Actinobacteria* and *Chloroflexi*, as well as depleting several *Acidobacteria* and *Deltaproteobacteria* in rice root-associated microbiota [27]. Similar dynamics of rhizosphere microbiomes in dry and rainy seasons demonstrated a change of dominants in different plant species from *Actinobacteria*, *Acidobacteria*, and *Bacillus* to *Proteobacteria* and *Bacteroidetes*, and suggested the existence of a “core microbiome” of drought-enriched taxa [28]. The effect of drought on the rhizosphere microbiome diversity is quite complex. Despite reports of decreasing rhizosphere microbiome diversity under water stress [28,29,30], there are reports of no effect or about the dependence of this effect on other parameters [31,32], which will be discussed in more detail in the discussion section. 

The current study is an extension and expansion of experimental work related to the effects of combined stresses caused by water deficit and cadmium (Cd) toxicity on two pea genotypes (line SGE and its Cd-tolerant mutant SGECd^t^) grown in a native soil [33]. Previously, we showed that the SGECd^t^ mutant possessed a Cd-insensitive phenotype, which manifested in better root and shoot growth and reduced physiological and biochemical reactions to Cd toxicity, but increased the accumulation of Cd in roots and shoots under hydroponic conditions [34,35,36,37,38]. This unique mutant retained the ability to counteract Cd-induced growth inhibition when cultivated in contaminated soil under a limited water supply [33]. Moreover, inoculation with nodule bacteria helped the mutant to form effective nitrogen-fixing symbiosis leading to the improvement in nitrogen nutrition and promoting plant growth [33,39], and thereby indicating that microorganisms play an important role in plant adaptation to stress conditions. Thus, the SGECdt mutant can be used as an original genetic model for studying the role of the plant genotype in response to abiotic stresses, as well as it can help in estimating plant-induced changes in the rhizosphere microbial communities subjected to adverse environmental conditions. Considering that the pot experiment [33] was carried out in soil, we propose that the plant stress tolerance modulates the rhizosphere microbial community, and the native microbiome could be a source for rhizosphere microbes contributing to the alleviation of environmental stresses.

To understand to what extent the rhizosphere microbiome can be affected by abiotic stresses and modulated by the plant genotypes varying in cadmium tolerance, we compared rhizosphere microbiome compositions of pea line SGE and its SGECd^t^ mutant in response to drought and/or Cd treatment. For this purpose, rhizosphere soil samples collected from the uninoculated plants during the aforementioned experiment [33] and incorporating the effects of three factors (drought, Cd toxicity, and plant genotype) in all possible combinations were used for the present study. The choice of factors was determined by the subject of the previous study [33]. The study [33] also demonstrated positive effects of symbiotic bacteria on plant tolerance to both Cd toxicity and water stress depending on the plant genotype. This indicates the significance of rhizosphere microorganisms for the adaptation of plants to stress factors.

The current work is an attempt to estimate how abiotic factors, namely water stress and cadmium stress, shape the rhizosphere microbiome of pea (wild-type line SGE and its Cd-tolerant mutant SGECd^t^). We show that the key stress factor was water stress, while cadmium and, to an even lesser degree, plant genotype induced smaller effects. This finding is possibly indirect evidence that SGECd^t^ overcomes cadmium stress at the expense of its own resources.

## 2. Results

### 2.1. The Taxonomic Structure of Rhizosphere Microbiomes

The taxonomic structures of all rhizosphere microbiome samples, according to the 16S rRNA data analysis, are presented in Figure 1a (phylum level) and Figure 1b (genus level). The prokaryotic community is dominated by bacteria constituting more than 99% of the microbiomes, and minor representatives of *Archaea* (0.1–0.6%) were detected in all samples analyzed. Representatives of *Proteobacteria* (mean abundance: 40.3%), *Actinobacteriota* (15.6%), unclassified taxa (28.2%), *Firmicutes* (3.2%), *Verrucomicrobiota* (3.2%), *Acidobacteriota* (2.9%), *Planctomycetota* (2.1%), *Gemmatimonadota* (1.9%), and *Bacteroidota* (1.5%) were dominants with mean abundances exceeding 1%.

From the taxonomic bargraphs, it can be concluded that the key factor shaping the rhizosphere community is most likely a watering regime. This is evident from the taxonomic picture of the genus rank (Figure 1b), which shows the difference between “water-stressed” and “normal” microbiomes. Higher abundances of *Mezorhizobium*, *Allorhizobium–Neorhizobium–Pararhizobium–Rhizobium*, and *Stenotrophomonas* are seen in the “normally watered” microbiomes, while higher abundances of some unclassified *Xantobacteraceae* genera are seen in the “water-stressed” microbiomes. The differences in the abundances at the phylum level are not that obvious, although there are some differences in the representation of actinobacteria in the normal and water-stressed samples. The difference between the Cd variants and pea genotypes is not obvious. The exact statistics of the balances based on the compositional data analysis approach are given below.

### 2.2. Analysis of Diversity Indices

The indices of the analysis of diversity (Figure 2) show a trend similar to that described above. Namely, statistically significant differences were detected only between samples with different watering regimes, and not between variants with and without cadmium supplement (only one case) or different plant genotypes. A notable observation is that the statistically significant increase in the diversity indices including Shannon, inverted Simpson, and phylogenetic diversity under water stress, but not in cases of cadmium supplement and plant genotype variants. A detailed analysis of this effect is given elsewhere since it refers to general ecology problems rather than to rhizosphere microbiome-related mechanisms of cadmium tolerance. When taking into account the main issue of this study, it is important to note here that the plant genotypes SGE and SGECdt had no statistically significant difference in the diversity of the rhizosphere microbiome under all factor combinations (including drought and cadmium supplement).

### 2.3. Analysis of β-Diversity

The effects of abiotic stresses and various plant genotypes, which are reflected in the taxonomy and diversity indices of the rhizosphere microbial community, are also seen in the analysis of β-diversity. This analysis was conducted after the phylogenetic isometric log-ratio transformation and ordination using the Euclidean metric (Figure 3a–c). The degree of separation of the rhizosphere microbiomes caused by water stress and cadmium supplement was more pronounced than in the case of various plant genotypes. This was confirmed by the data of the permutation test using Adonis2 (Table 1). The data confirm a statistically significant effect of the watering regime and cadmium supplement but not the plant genotype. The results of the Adonis2 test also show a lesser but still significant interaction of the cadmium supplement with watering, and a cadmium supplement interaction with the plant genotype.

### 2.4. Compositional Data Analysis

The final part of this study was gaining an understanding of the particular taxa dynamics in response to abiotic stresses and plant genotype variants. Unlike traditional statistical approaches to differentially presented taxa, we took the compositional nature of microbial communities’ taxonomic structure data into consideration. The main characteristic of the compositional data analysis (CoDA) is the inclusion of operations with balances (that is, the log ratio of the geometric means of frequencies for subtrees descending from a particular node in the phylogenetic tree) rather than with the frequencies themselves. Two main measures of these balances are the phylogenetic depths in the tree and the corresponding log ratio value. For example, the deepest “watering” balance b459, the “cadmium” balance b685, and the “pea” balance b459 affected 121, 5, and 28 ASV, respectively. Figure 4 shows a binary phylogenetic tree with the taxonomic balances related to the watering regime, cadmium supplement, and plant genotype. The data presented demonstrate that the most powerful factor affecting the rhizosphere community is the watering regime, showing the deepest phylogenetic balances with the highest log-ratio values. For example, the deepest identified balance, #459, between the order *Burkholderiales* and an unclassified *Alcaligenaceae* genus demonstrates log-ratio values of about −2.5 (normal watering) and +2.5 (water deficient). Contrary to the phylogenetically deep and powerful balances mediated by the watering regime, numerous balances related to the plant genotype are shallow and very weak, with the log-ratio values not exceeding ±1.0. Cadmium-related balances had an intermediate position in regard to the log ratio (but not phylogenetic depths).

## 3. Discussion

It was suggested that the taxonomic structures of the plant rhizosphere microbiomes have common features expressed as the enrichment of microorganisms belonging to *Proteobacteria* and *Actinobacteria* [40,41]. The domination of *Proteobacteria* and the underrepresentation of other common soil phyla are probably conditioned by a specific rhizosphere environment rich in easily consumable components of plant root exudates (carbohydrates, organic and amino acids). This leads to an increase in the proportion of copiotrophes, many of which belong to *Proteobacteria*. Although the main method applied here for determining the differences in taxa abundances was the use of a compositional data-based approach, something notable can be speculated from the taxonomic barplots. The apparent prevalence of proteobacterial genera from the *Mezorhizobium*, *Rhizobium*, *Allorhizobium*, *Pararhizobium*, and *Neorhizobium* complex (Figure 1b) in the rhizosphere of the normally watered plants was likely related to water stress modulating the rhizobial component of the pea rhizosphere. It is important to note that in all these experimental variants, rhizobial inoculation was not applied. The detected nodule formation was due to spontaneous inoculation by resident soil rhizobia, and the number of nodules produced was about four times lower than in inoculated variants, but no statistically significant difference between the plant genotypes with all watering and cadmium supplement variants was detected [33]. On the other hand, the separation of the rhizosphere microbiome fraction was carried out with all precautions in order to avoid damage to the nodules. This is why we believe that the rhizobial pool detected in the 16S rRNA gene libraries is not that of the nodule; rather, it probably represents the rhizosphere rhizobia recruited from the soil by the legume exudation of organic compounds, including flavonoids. In this regard, a decrease in the rhizobial fraction under conditions of water stress is of particular interest, since it might be related to different strategies of organisms, namely “oligotrophy” and “copiotrophy” [42,43], or to the fate of copiotrophes under abiotic stress. Since the study focus was on the compositionality of the data, it must be established that it cannot be strictly said that the stress led to a decrease in the absolute abundance of this group. Rather, only a decrease in the group’s relative abundance was observed, with increases in the abundances of other groups as one possible cause. When assessing and discussing changes in the taxonomic structure in our study, the concept of balances was applied rather than a comparison of the abundances in the groups.

Another important point is the analysis of changes in the diversity indices. The data presented in Figure 2 demonstrate that the main factor affecting the microbial diversity in the rhizosphere was the watering regime. None of the indices representing different aspects of diversity and their combinations (evenness, richness, and phylogeny) demonstrate significant differences between the two pea genotypes. Cadmium stress had an intermediate position. Regarding the index changes, the long-speculated question of how soil quality (fertility, health, and sustainability) [44] is related to microbial diversity cannot be ignored. With the introduction of molecular methods into practice, the real size and diversity of soil microbiomes became clear, and this discussion point returned to the agenda [45,46,47]. Now, it is generally accepted that the soil microbiome delivers multiple ecosystem services and functions, and that its diversity is related to the soil quality. A parallel discussion exists on the effect of stress factors on the diversity of the microbial community. A common claim in such studies, which include high-throughput sequencing data, is the recognition that stress results in a decrease in alpha diversity [9]. However, in the case of water stress, the dynamics of diversity indices are not so obvious. Despite the fact that there are reports of the dynamics of rhizosphere microbiomes under water stress conditions, which show a decrease in diversity indices [28,29,30], there are also reports of the absence of such dynamics [31] or their dependence on the stage of plant development [32]. It also should be noted that the authors of these studies have not adhered to a uniform approach to assessing diversity (abundance, evenness, richness, etc.), and this complicates the overall assessment. In this particular case, we present a clear result: drought stress led to statistically significant increases in all alpha diversity indices related to evenness, richness, and phylogenetic diversity. This effect is likely a result of complex interactions in the rhizosphere, including: (1) increases in a small number of particular copiotrophic taxa in the rhizosphere with normal watering; (2) the suppression of these taxa by water stress. A detailed analysis of these effects needs to be further studied in more detail.

The results of the beta diversity analysis (Figure 3 and Table 1) are in line with the above and further discussed results. The main factor shaping the rhizosphere microbiome diversity was the watering regime, whereas the role of the plant genotype was insignificant. It is worth mentioning that in the beta diversity ordination, there appears to be a deeper differentiation by cadmium as compared to that of the water regime. To explain this not high but statistically significant fluctuation, it must be taken into account that the use of “compositional” statistics for the ordination and the PERMANOVA test was performed using a table of balances, according to this work’s approach, without taking into account the taxa abundances. The reason for this is likely in the specificity of ordination based on the balances. A significant challenge in this study was a possible decrease in the mobility of cadmium ions with the water stress, and, as a consequence, a decrease in the absorption of cadmium by the plant, with further difficult-to-control changes in the rhizosphere microbiome. It is difficult to investigate the role of individual factors and tolerance mechanisms of plants in the case of the combinations of various stress factors. This problem has been discussed by several authors [48,49]. However, the results of the original work [33] demonstrated that pea plants under water stress continued to accumulate cadmium (although to a lesser extent) and the differential accumulation in SGE and SGECd^t^ still took place. The results of the rhizosphere microbiome analysis demonstrate that there was clear differentiation between the cadmium-affected microbiomes in the cases of both normal watering and water stress. We believe that a relatively weak interaction between the factors detected by the PERMANOVA test, namely the cadmium–water regime and cadmium–genotype, can be left out of the discussion because the design of the experiment (three factors in all possible combinations) allows us to divide the same dataset in three different ways (leaving the two factors equally represented in each of the divisions) and obtain the “integral” effects of each factor without going into the finer details of the interactions between the factors.

The final step in this analysis is the listing of significant balances presented in Figure 4. An important feature of the present work is the use of statistics, taking into consideration the compositional nature of the microbial communities’ taxonomic structure data. In such data sets, the proportions of the taxa identified in libraries are not independent: a change in the quantity of one component leads to a change in the proportions of others, even if their absolute quantities remain unchanged. Therefore, the analysis of proportions requires unconventional statistical methods under specific compositional assumptions [50]. The main feature of the compositional data analysis (CoDA) is different types of isometric log-ratio transformations [51]. This approach is especially important when high-throughput sequencing data (which are compositional) need to be analyzed [52,53]. In this work, we used a binary tree built on the basis of the phylogeny [54]. There are three important parameters of a balance, namely the taxonomy, the phylogenetic depth, and the log-ratio value. It must be noted that the binary tree presented in Figure 4 reflects only a hierarchy and not real distances, and based on this, we can estimate how many lowest-ranking taxa were affected by this balance. The deepest and most “powerful” balance was the “watering” balance b459, which is in agreement with the above-mentioned findings, indicating that the watering regime was the main factor affecting the rhizosphere microbiome in this experiment. In general, the water and cadmium stresses were distinguished by a small number of relatively powerful balances (log ratio of about −3; +3), while the balances related to the plant genotype were more numerous but very weak (log ratio of −0.8; +0.8). From a taxonomic point of view, it makes sense to discuss only those balances “operating” with different taxa. Such balances are mostly water stress-related. For the high-level balance b34, it can be seen that representatives of the order *Burkholderiales* displaced representatives of the family *Alcaligenaceae* (separate family of *Burkholderiales*). It is known that many representatives of *Burkholderiales* are PGPR bacteria [55] and some authors have reported that their presence increased with a decrease in watering [56,57]. Other authors have reported otherwise; it increased with watering [58]. Some representatives of *Alcaligenaceae* are reported as bacteria responsive to drought stress [59] and as PGPR, reducing water stress for plants [60]. Another high-level balance, b459, shows displacement of Micrococcales by a large clade of *Actinobacteria*. Many representatives of *Actinobacteria* are drought-tolerant and inhabit arid areas [61]. Among them, there are the PGPR bacteria that mitigate water stress for plants [62,63], while *Micrococcales* are known as bacteria that inhabit soil as well as water [64]. Balance b824 featured the displacement of some representatives of *Pedosphaeraceae* by other representatives of its clade. These bacteria are active members of biogeochemical cycling [65] that are resistant to the presence of cadmium [66]. Cadmium and pea genotype-related balances were located within the same taxon in all cases and did not give grounds for discussion of the taxonomic features of these balances. It is worth special mention that there are no taxa in the list of balances, the changes in which appear to be significant in Figure 1 (*Mezorhizobium*, *Allorhizobium–Neorhizobium–Pararhizobium–Rhizobium*, *Stenotrophomonas* and unclassified genera from *Xantobacteraceae* family). However, the main task of compositional statistics is to identify taxa that demonstrate the dynamics of absolute abundances, but not relative ones (proportions). In this light, the discrepancies between traditional and compositional approaches (in taxonomic details) are not surprising, especially since both approaches are similar in the main message (in the hierarchy of factors).

The main conclusion from this part of the study is that the most powerful factor affecting the taxonomic balances was water stress, and the weakest factor was plant genotype since it demonstrated very weak transformation of the taxonomic structure of the rhizosphere microbiomes, especially in terms of the log-ratio values. This is consistent with the entire set of data obtained in the experiment and can be taken as the general conclusion of this work. Nevertheless, the nature of the plant mutant SGECd^t^ and its relationship to the findings of this work merit further discussion. The problem is that most works in which the role of the plant genotype in the formation of the rhizosphere microbiome has been clearly demonstrated have dealt with different plant genotypes [5]. Meanwhile, in the present work, the focus of study was on two extremely close pea genotypes with minimal genetic differences due to an induced mutation (until today, not localized). We believe that negligible differences between the rhizosphere microbiomes of SGE and SGECd^t^ pea genotypes are indirect evidence of the strong similarity of the two genotypes in all of the features defining the specificity of rhizosphere environment (root exudation, immune status, etc.). Another point is that these minimal differences indicate that the rhizosphere microbiome is unlikely to take part in the cadmium resistance of SGECd^t^, and its contribution is approximately equal in both pea genotypes. Ultimately, the pea mutant overcomes the stress at the expense of its own resources.

## 4. Materials and Methods

### 4.1. Plant Growth Conditions

A detailed description of the plant growth conditions was presented earlier [33]. In brief, pea genotypes SGE and its Cd-tolerant mutant SGECd^t^ were grown in a polyethylene greenhouse under natural illumination and temperature (June–July, St. Petersburg) in pots containing 2 kg of native sod–podzolic soil. Two watering regimes, 40% WHC (water stress) and 80% WHC (sufficiently watered), were imposed with evapotranspirational losses replenished every day. Part of the soil was supplemented with 30 mg Cd kg^−1^ as CdCl_2_. All 8 possible variants, combining two watering regimes, two treatments with Cd, and two pea genotypes, were prepared in quadruple replication. The plants were cultivated for 45 days. At harvest, the roots were removed from the soil and collected for further rhizosphere soil fraction analysis.

### 4.2. Rhizosphere Soil Fraction Separation and DNA Isolation

The roots were freed from the loosely attached soil particles by shaking in empty sterile plastic bags, placed in 50 mL Falcon tubes with 25 mL sterile 0.9% solution of NaCl, and softly shaken for 5 min to avoid the destruction of root tissues and nodules. Then, the suspensions were centrifuged by 5000× *g* (6526 rpm for the Eppendorf rotor F-35-6-30) for 5 min, and the precipitates of the rhizosphere soil fraction, along with microorganisms were immediately frozen at −20 °C and subsequently used for DNA isolation. A total of 4 independent rhizosphere soil samples were taken for each of the 8 experimental variants.

### 4.3. Preparation of 16. S rRNA Library and High-Throughput Sequencing

For DNA extraction, MACHEREY-NAGEL NucleoSpin Soil kit (MACHEREY-NAGEL, Düren Germany) was used. The initial lysis was performed with 700 μL SL1 buffer with 150 μL of enhancer. Further steps were performed in accordance with the manufacturer’s instructions. The final elution was in 50 μL of elution buffer.

The library preparation (32 libraries in total) for sequencing included the amplification of the target fragment of the variable region V4 of the 16S rRNA gene using universal primers (515F-GTGCCAGCMGCCGCGGTAA/806R-GGACTACVSGGGTATCTAAT) [67] together with linkers and unique barcodes. PCR was performed on a T100 Thermal Cycler (BIO-RAD Laboratories, Heracles, CA, USA) in 15 μL of a reaction mixture containing 0.5 units of Q5^®^ High-Fidelity DNA Polymerase (New England BioLabs, Ipswich, MA, USA), 1X Q5 Reaction Buffer, 5 pM of each primer, 2 mM dNTP (LifeTechnologies, Carlsbad, CA, USA), and 1–5 ng DNA template. The PCR program included the stage of denaturation at 94 °C—1 min, amplification of the product for 35 cycles (94 °C—30 s, 50 °C—30 s, 72 °C—30 s), and final elongation at 72 °C—3 min. Further sample preparation and sequencing were carried out in accordance with Illumina protocol (“16S Metagenomic Sequencing Library Preparation”) on an Illumina MiSeq device (Illumina Inc., San Diego, CA, USA) using a MiSeq Reagent Kit v3 (600 cycles) with pair-end reading (2 × 300 b) (Illumina Inc., USA).

### 4.4. Data Analysis

The initial data processing, including demultiplexing and adapter trimming, was performed using Illumina Software (Illumina, USA). For further denoising, merging reads, inferring amplicon sequence variants (ASV), and removing chimera, the dada2 [68] package in R software was used. For the taxonomic classification of ASVs, the DECIPHER [69] package was used. To train the classifier, we extracted v4 rRNA fragments from the SILVA database (138 release) [70] records and used them as a training set for the function LearnTaxa (DECHIPHER). The classification of ASVs was performed with the IdTaxa function with a confidence threshold equal to 70. To construct a phylogenetic tree, the SEPP fragment insertion algorithm implemented in the QIIME2 plugin was used [71]. The biome table rarefaction procedure (phyloseq [72], picante [73]) was used to assess alpha diversity. Ggpubr packages [74] were used to build violin plots with the reliability of differences. For p-adjusted correction, the Bahjamini–Hochberg method was used. The QIIME package [75] was used to build barplots for the taxonomy.

### 4.5. Compositional Data Analysis

Using the compositional data approach, we used log-normalized balances. Balances are nodes within the binary phylogenetic tree, and their value is the log ratio of the geometric means of the parts (relative abundances for the subset of leaves) descending from a particular node. This coordinate system is the orthonormal basis in which conventional statistical methods work without constraints typical for compositional data [52,54].

To eliminate the “noise” in the data, namely a large number of zeros, phylotypes were removed from the analysis. The representation of the phylotypes was at least 3% in less than 20% of the samples, after which the additive elimination of zeros was carried out (adding 1 to all absolute numbers). To transform the data, a phylogenetic isometric log-ratio transform implemented in the philr package [54] was used. To select features (balances), we used cross-validation with lasso regression in the glmnet package [76]. The distance between the samples was estimated using the Euclidean metric for the transformed data. The ordination was constructed using PCoA (the phyloseq package). The construction of box plots and the visualization with a phylogenetic tree were carried out using the ggplot2 [77] and ggtree [78] packages, respectively. To assess the significance of the effect of factors such as watering, cadmium supplement, and plant genotype, the adonis2 (PERMANOVA) [79] test on the matrix of Euclidian distances was performed.

## Figures and Tables

**Figure 1 plants-11-03013-f001:**
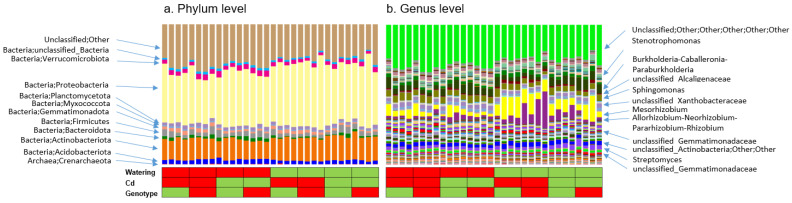
Taxonomic structure of rhizosphere microbiomes inferred from 16S rRNA library high-throughput sequencing: (**a**) phylum level, (**b**) genus level. Hereafter, the same scheme for designating factors is followed: green—normal watering, no cadmium, original pea genotype SGE, red—water stress, cadmium additions, SGECd^t^ mutant.

**Figure 2 plants-11-03013-f002:**
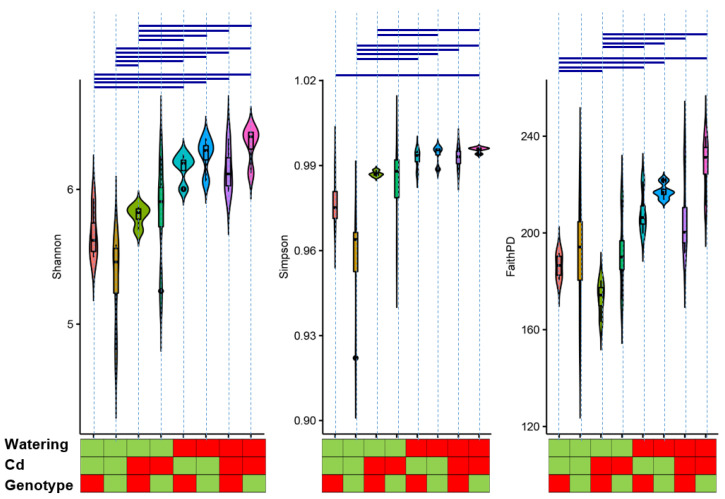
Diversity indices in rhizosphere microbiomes of SGE and SGECd^t^ plant genotypes under different combinations of abiotic stress factors. For the color scheme, see caption of Figure 1. Blue lines connect values with significant differences.

**Figure 3 plants-11-03013-f003:**
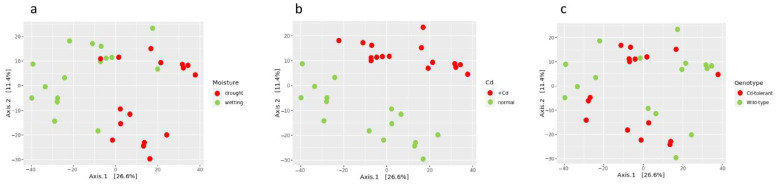
β-diversity of rhizosphere microbiomes after ilr transformation and ordination using a Euclidian metric colored according to watering regime (**a**), cadmium supplement (**b**), and plant genotype (**c**). Axes are percentages of variance explained. For the color scheme, see caption of Figure 1.

**Figure 4 plants-11-03013-f004:**
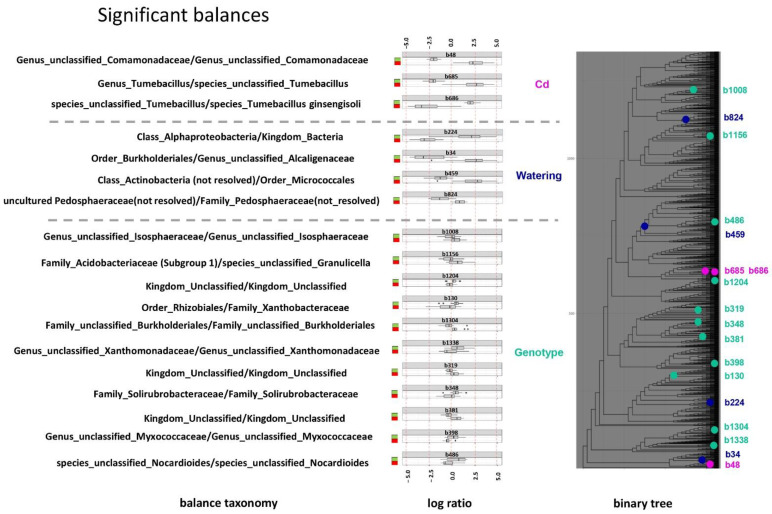
Statistically significant taxonomic balances related to watering regime, cadmium supplement, and plant genotype. Taxonomy of balances reflects the last identified rank. Binary tree does not reflect real genetic distances, only hierarchy. For the color scheme of log ratios, see caption of Figure 1.

**Table 1 plants-11-03013-t001:** Results of PERMANOVA test of factor significance.

	Df	Sum of Sqs	R^2^	F	Pr (>F)
Cd	1	7782	0.13323	5.9442	***
Watering	1	9186	0.15727	7.0169	***
Genotype	1	1733	0.02967	1.3238	
Cd/watering	1	2573	0.04405	1.9652	*
Cd/genotype	1	2826	0.04837	2.1583	*
Watering/genotype	1	1583	0.02710	1.2092	
Residual	25	32,729	0.56032		
Total	31	58,412	1.00000		

Signif. Codes: 0 ‘***’ 0.01 ‘*’.

## Data Availability

Raw sequence data are available at SRA with accession number PRJNA838476.

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
