# Peer review of "Water Stress, Cadmium, and Plant Genotype Modulate the Rhizosphere Microbiome of Pisum sativum L."

_plants, 2022, doi:10.3390/plants11223013_

Round 1

Reviewer 1 Report

General comments

The manuscript “Water deficit, cadmium and plant genotype modulate rhizosphere microbiome of Pisum sativum L.” by Kichko et al., reports the observed substantial effects of the limiting moisture and/or toxic cadmium compared to the insignificant influence of plant genotype, on the growth and rhizosphere (microbiome) biodiversity of Pisum sativum L. – while the rhizospheric biodiversity plays role in the adaptation of these plants to the environmental stressors of drought and toxic metal concentration. While the abstract provides the scientific research hypothesis, methodology and unconventional statistical techniques, no plain language take-home message wraps up the abstract –Though, the authors provide some message at the end of the introduction (which is an unconventional place to provide a take-home message), or between lines 297-300It. The introduction is rather long, though interesting. Authors may derive the TAKE-HOME message from lines 297-300 of the discussion section.

I would be much interested to know the modulation threshold level of the Pisum sativum L. rhizosphere communities.

I am a bit confused about the use of the term “water deficit”, which authors might mean “water status in plant low enough to affect normal plant functioning” – compared to the simple hydrological term “water deficit” is equal to Precipitation minus Evapotranspiration. Therefore, authors may use the term moisture stress, water stress or low water potential, which are closer to the plant physiological terminology. At the minimum, clearly defining what water deficit they mean is required in one of the early paragraphs of the introduction.

Unlike most research works, I believe, this research work is unique and novel in a way that it evaluates the individual and complicated combined interactive effects of abiotic (drought and toxic metal) and biotic (genotype) factors on the rhizosphere microbiome biodiversity – the focus is a modern topic, which I believe aligns with the interest of the readers of the “plants” and specifically fits the special issue “Adaptation of Mutualistic Plant-Microbe Systems to Abiotic Stresses”. The write-up merits to be published in plants after the reviewer’s comments are substantially addressed.

Specific comments

Line 35: correct grammar used in “An important factors that affects”. This sentence may imply that all heavy metals are always at toxic levels and are a source of anthropogenic pollution – rewrite or correct.

Line 37-38: correct grammar – add “is”

 Line 43: correct to consumable

Line 50: may help the plant

Line 95: I am unsure how the soil collected for pots was unsterile.

Fig. 1: looks good and orderly – but I do not frequently work on high throughput sequencing. However, correct to “throughput”.

Line 206-207: pretty confusing – what do the authors mean here and what is the point?

Line 208: correct grammar

Lines 206-211: Not sure how the authors want to set the stage for discussion – unclear.

Lines 321-322: how did researchers measure evapotranspiration? Belimov et al. 2019 say – “The plants were cultivated for 45 DAP, with whole plant transpiration rate monitored gravimetrically from 20 to 45 DAP, and corrected for soil evaporation (using pots without plants) for both watering regimes.” My question is were the pots of PVC or any other material?

Line 331: what is centrifuged by 5000 g?

Line 348: authors need to be consistent in the use of °C (degree sign before or after C) here and throughout the manuscript.

Line 250-252: Authors provided only one reference here Rocca et al. 2018; they need to provide several references of parallel reports on the effect of stress factors on the rhizosphere microbial diversity.

Author Response

The authors are very grateful to the reviewer for critical reading the manuscript and valuable comments. We tried to address all of your comments.

The authors provide some message at the end of the introduction (which is an unconventional place to provide a take-home message), or between lines 297-300It. The introduction is rather long, though interesting. Authors may derive the TAKE-HOME message from lines 297-300 of the discussion section.

Response: The last part of the abstract has been modified according to the reviewer's suggestion. lines 22-25

The introduction is rather long, though interesting.

Response: According to yours and the second reviewers’ suggestion, we corrected and shortened the introduction and add some references to make it more logical.  

I would be much interested to know the modulation threshold level of the Pisum sativum L. rhizosphere communities.

Response: I am not sure that I understand the reviewer’s question correctly. If the question is to what extent the rhizosphere community can change its structure in response to abiotic and biotic factors, the answer could be like this. Root exudation is the basic mechanism of recruiting microorganisms from the soil microbiome. If we take as a measure of the modification level of the soil microbiome, the distance between soil and rhizosphere microbiome (to say Bray-Curtis or UniFrac distances), we will expect the maximal difference in the case of physiologically healthy plant with unlimited access to nutrients and water. In the case of deficit of something, the structure of rhizosphere community will be closer to the parental soil community. On the other hand, the stressed plants could respond by changes in root exudation to activate interactions with beneficial microorganisms as a mechasnism for adaptation to this stressful conditions. For example, we showed previously that Cd sensitive pea genotypes rely more actively on symbiotic microbes (mycorrhizal fungi and rhizobia) as compared to Cd tolerant genotypes (Belimov et al., Genetic variability in tolerance to cadmium and accumulation of heavy metals in pea (Pisum sativum L.). Euphytica, 2003, 131, 25-35, https://doi.org/10.1023/A:1023048408148. Indeed, this question is very interesting and important but very complicated. We addressed this problem to some extent in our other works.

Please, see: https://link.springer.com/article/10.1134/S002626172001018X; https://www.mdpi.com/2076-2607/9/11/2339

Specific comments

Line 35: correct grammar used in “An important factors that affects”. This sentence may imply that all heavy metals are always at toxic levels and are a source of anthropogenic pollution – rewrite or correct.

Response: Corrected.

Line 37-38: correct grammar – add “is”

Response: Corrected.

Line 43: correct to consumable

Response: Corrected.

Line 50: may help the plant

Response: Corrected.

Line 95: I am unsure how the soil collected for pots was unsterile.

Response: Corrected. We meant “unsterilized”, but in the final version change it for “native”.

Fig. 1: looks good and orderly – but I do not frequently work on high throughput sequencing. However, correct to “throughput”.

Response: Corrected.

Line 206-207: pretty confusing – what do the authors mean here and what is the point?

Response: The purpose of this paragraph is to show that the taxonomic structure of rhizosphere microbiomes is quite common, since it is dominated by proteobacteria and actinobacteria. In particular, the predominance of proteobacteria is a distinctive feature of rhizosphere microbiomes, since they are responsive to easily consumable components of root exudates. To avoid confusing, we modified this part of the text.

Line 208: correct grammar

Response: Corrected.

Lines 206-211: Not sure how the authors want to set the stage for discussion – unclear.

Response: I hope that after correction the stage became clearer. Three first sentences are just presentation of the main characters typical for the rhizosphere community. The setting of the stage for discussion starts from the fourth sentence “Although the main method applied…”. We believe that this order of presentation is important, since the result of exposure to stress factors in this case consists of two components, namely the recruitment of some taxa from the soil to the rhizosphere, and then the impact of stress factors on them. We corrected the text, see lines 219-225

Lines 321-322: how did researchers measure evapotranspiration? Belimov et al. 2019 say – “The plants were cultivated for 45 DAP, with whole plant transpiration rate monitored gravimetrically from 20 to 45 DAP, and corrected for soil evaporation (using pots without plants) for both watering regimes.” My question is were the pots of PVC or any other material?

Response: The pos were enameled metal without holes in the bottom. To aerate the roots, glass tubes were inserted into the soil almost to the bottom of the pot, which provided air access to the roots (aeration) and were also used for watering the plants. Such a system allowed water, cadmium and nutrients to remain in the pot. Water could only be transpired by plants and evaporated from the soil surface. Evaporation from the soil surface was taken into account using vessels without plants. All vessels were weighed daily. The amount of water transpired by plants (evaporation by plants) was calculated as the difference in the weight of pots with and without plants. This is very simple and common technique applied by many researchers. Therefore we didn’t describe these details in the manuscript.

Line 331: what is centrifuged by 5000 g?

Response: Since when using rotors of different diameters rcf (g) can differ at the same rotational speed (rpm), we indicated the rcf value. For clarity, we have also indicated the type of rotor and the speed. We corrected this text according to the rules of Plants journal as 5000× g.

Line 348: authors need to be consistent in the use of °C (degree sign before or after C) here and throughout the manuscript.

Response: Corrected.

Line 250-252: Authors provided only one reference here Rocca et al. 2018; they need to provide several references of parallel reports on the effect of stress factors on the rhizosphere microbial diversity.

Response: References and some discussions have been added.

Reviewer 2 Report

Kichko et al, reported the biodiversity, occurrence, and role of microbiomes present in the rhizospheric zone of pea plants exposed to drought and cadmium toxicity. This study is quite interesting, especially the use of mutant genotype of pea, and can add more information to the scientific literature with new investigations. However, a few concerns could be addressed before further processing this article. These concerns are listed below:

(i)            Why authors only focused on drought and cadmium stress? Other abiotic stresses also possess negative impacts on plant growth and soil properties. Need an appropriate answer.

(ii)          Both stresses (drought and cadmium) are applied together to assess the response of pea rhizospheric regions. Generally, nutrients/ions/metal ions from soils face limited mobility and uptake under drought conditions. How authors can correlate the impact of both stresses together? Justify.

(iii)        M & M section is missing the method of Cd application and how and why they select only 30mg/Kg concentration.

(iv)        Authors are advised to write their full name as their first appearance. For example Line 13 “SGE and its Cd-tolerant mutant SGECd”. Carefully remove such kind of errors in the whole manuscript.

(v)          Introduction section is too descriptive. Authors are suggested to remove duplication of sentences and keep relevant literature.

(vi)        How do authors set 40% and 80% water holding capacity? need description in the manuscript as well.

(vii)      Why authors had not focused on or analyzed plants related parameters while doing soil microbial analysis?

(viii)    Carefully check the reference style according to the Journal style.

Overall, this study could be improved before further process. However, the final decision will be with the Editor.

Author Response

The authors are very grateful to the reviewer for critical reading the manuscript and valuable comments. We tried to address all of your comments.

(i)            Why authors only focused on drought and cadmium stress? Other abiotic stresses also possess negative impacts on plant growth and soil properties. Need an appropriate answer.

Response: The choice of factors is determined by the subject of the previous study (Belimov et al., 2019) and the objects of which (soil samples) we used for this analysis. In that study the effects of cadmium or/and drought on pea genotypes differing in Cd tolerance (line SGE and its Cd-tolerant mutant SGECdt) were studied. See Belimov et al., Genetic variability in tolerance to cadmium and accumulation of heavy metals in pea (Pisum sativum L.). Euphytica, 2003, 131, 25-35. https://doi.org/10.1023/A:1023048408148. Growth response, and accumulation of cadmium were also involved in that report. Drought is one of the most important environmental stresses causing decrease in yield of agricultural crops, including pea. Including additional stress factor should make the story too much complicated for analysis. We modyfied patagraph on lines 104-114 trying to address your comment.

(ii)          Both stresses (drought and cadmium) are applied together to assess the response of pea rhizospheric regions. Generally, nutrients/ions/metal ions from soils face limited mobility and uptake under drought conditions. How authors can correlate the impact of both stresses together? Justify.

Response:

Indeed, under natural conditions the plans very often subjected to combinations of various stress factors. Therefore, it is difficult to investigate the role of individual factors and tolerance mechanisms of plants in situ. This problem has been discussed by several authors. To correlate the impact of both stresses together we applied negative (uncontaminated soil, well-watered soil) and positive (contaminated well-watered soil, uncontaminated droughty soil) control treatments. Despite the drought, in this experiment, cadmium continues to accumulate in pea plants, although to a lesser extent and the differential accumulation in SGE and SGECdt still takes place (Belimov et al., 2019, Fig. 3). Results of our experiment with microbiomes, especially β-diversity (Fig 3 a and b) demonstrates clear difference between effects of Cd and water stresses. Cadmium affected microbiomes are separated in all water conditions and water affected microbiomes do the same in cadmium variants. But indeed there is a tiny “intermixing” zone of cadmium stressed variants under different water conditions (Fig 3 a), probably reflected in PERMANOVA test demonstrating smaller, but statistically significant interaction between cadmium and water effects (Table 1). However, in this work we focus on the integral effects of water and cadmium stress, because the design of variants with three factors in all possible combinations give us the opportunity to divide the same dataset in three different ways (leaving the two factors equally represented in each of the divisions) and obtain the “integral” effects of each factor without going into the finer details of interactions between factors. We expanded the discussion and added new references in the lines 285-300.

(iii)        M & M section is missing the method of Cd application and how and why they select only 30mg/Kg concentration.

Response: The method of Cd application has beed described in detail in our previous publication where the effects of stress factors on plant growth and uptake of Cd and nutrients were present. Please, see: Belimov et al., 2019. doi:10.1016/J.ENVEXPBOT.2019.103859.

(iv)        Authors are advised to write their full name as their first appearance. For example Line 13 “SGE and its Cd-tolerant mutant SGECd”. Carefully remove such kind of errors in the whole manuscript.

Response: The SGE and SGECdt are the names of pea line and its mutant, respectively. These are proper names and the abbreviation is not deciphered.

(v)          Introduction section is too descriptive. Authors are suggested to remove duplication of sentences and keep relevant literature.

Response: According to yours and the first reviewers’ suggestion, we corrected and shortened the introduction and add some references to make it more logical. 

(vi)        How do authors set 40% and 80% water holding capacity? need description in the manuscript as well.

Response: Maintaining the water holding capacity was done by weighing the pots daily and adding water until they are of the same weight. The weight of vessels with moistened soil was calculated taking into account the moisture content of the soil, which is expressed in water holding capacity. This has beed described in detail in our previous publication where the effects of stress factors on plant growth and uptake of Cd and nutrients were present in Belimov et al., 2019. (doi:10.1016/J.ENVEXPBOT.2019.103859).

(vii)      Why authors had not focused on or analyzed plants related parameters while doing soil microbial analysis?

Response: All plants related parameters where reported in Belimov et al., 2019 (doi:10.1016/J.ENVEXPBOT.2019.103859. This work is an extension of the original work for studying the rhizosphere microbiome as an additional plant related (and plant driven) “organ” which helps plant to cope with abiotic stresses. In this work we use the rhizosphere soils sampled in that study.

(viii)    Carefully check the reference style according to the Journal style.

Response: Now we used the Mendeley for managing references and hope it helps to correct all problems.